# DEEP GENERATIVE PRIOR FOR FIRST ORDER INVERSE OPTIMIZATION

## ABSTRACT

Inverse design aims to recover system parameters from observed responses, a central challenge in domains such as semiconductor manufacturing, structural engineering, materials science, and fluid dynamics. The absence of explicit mathematical formulations in many systems complicates this task and prevents the use of standard first-order optimization methods. Existing approaches, such as generative models and Bayesian optimization, mitigate these challenges but face notable limitations: generative models often require high-fidelity paired data, while Bayesian optimization depends heavily on surrogate models, leading to scalability issues, sensitivity to priors, and vulnerability to noise.

We introduce **Deep Generative Prior (DGP)**, a new framework that enables first-order, gradient-based inverse optimization with surrogate machine learning models. Formally, DGP constrains the optimization of design parameters through a pretrained prior $G(q)$, such that gradients are propagated via the surrogate forward model $F$, i.e., $\nabla_q \mathcal{L}(F(G(q)), u)$, which enforces optimization along the data manifold induced by $G$. By leveraging pretrained Neural Operators as auxiliary priors, DGP enables stable and effective gradient flow through complex surrogate models.

We validate DGP on diverse and challenging inverse design tasks, including 2D Darcy flow (**standard**), 2D Navier–Stokes fluid dynamics (**ill-posed**), and semiconductor lithography inverse problems (**ill-posed** and **out-of-distribution solutions**). Across these domains, DGP consistently achieves higher solution quality and efficiency compared to existing methods.

## 1 INTRODUCTION

Inverse design optimization addresses the challenge of identifying system parameters or objectives ($a$) from observed solutions ($u^*$), making it fundamental across multiple disciplines. For example, in structural engineering, inverse design is used to infer the location and extent of damage based on sensor data. In chip design, inverse lithography is critical for optimizing mask patterns to improve manufacturability. Similarly, in materials science, inverse design identifies atomic or molecular structures that achieve desired properties such as thermal conductivity or elasticity. Aerodynamics also relies on inverse optimization for airfoil design. These examples highlight the wide-reaching impact of inverse design and its inherent ill-posedness, where multiple solutions may satisfy the same observations.

**Limitations of MCMC and Surrogate-Based Inference.** Solving inverse problems often relies on surrogate models that approximate the forward system. Classical approaches such as Bayesian optimization or Markov Chain Monte Carlo (MCMC) Cotter et al. (2013) repeatedly query the surrogate. While gradient-based MCMC variants, such as NUTS, can exploit the differentiable nature of neural surrogates, they remain computationally intensive for high-dimensional or ill-posed problems. Moreover, even with gradient-informed sampling, MCMC requires careful tuning of priors and step sizes to explore the solution space effectively. In contrast, data-driven surrogates like the Fourier Neural Operator (FNO) provide orders-of-magnitude speedup Li et al. (2021). Our proposed Deep Generative Physics prior (DGP) leverages both the differentiability of FNOs and a learned generative manifold to efficiently guide first-order optimization. This allows DGP to rapidly produce multiple physically plausible solutions while mitigating unphysical outputs, which remains challenging for standard or gradient-based MCMC approaches.

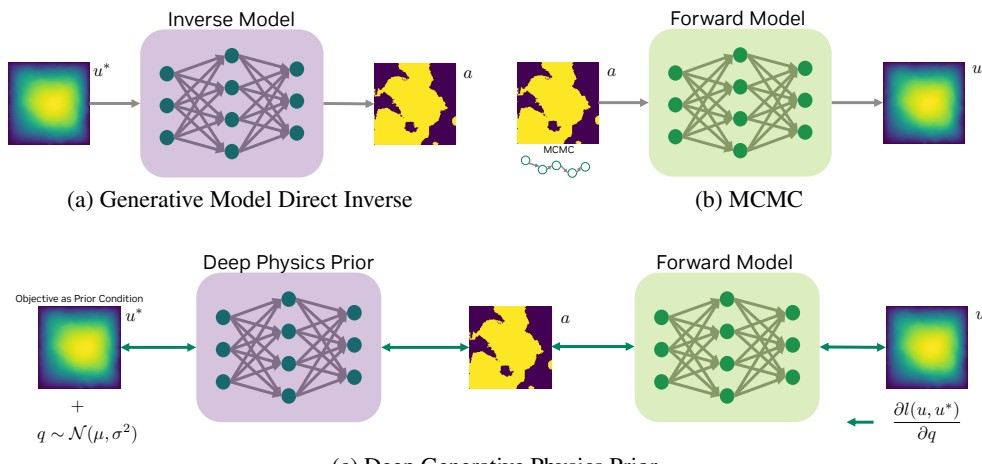

Figure 1: Comparison of inverse design optimization schemes. (a) Data-driven generative model mapping objectives to parameters. (b) MCMC sampling using a pretrained surrogate. (c) First-order optimization of system parameters constrained by a deep generative prior (DGP).

**Limitations of Inverse Operator and Direct Generative Models.** Generative models can directly predict inverse solutions Yang et al. (2020); Huang et al. (2024); Long & Zhe (2024); Yang & Ren (2024). While computationally efficient, their performance is highly sensitive to the underlying data distribution. In cases where the training data is multimodal or contains mixed distributions, these models often produce averaged or biased solutions that are noisy or physically implausible, requiring additional fine-tuning or optimization. Recent advances in diffusion-based generative models, such as Compositional Generative Inverse Design Wu et al. (2024), enhance robustness to multimodal distributions and enable compositional design, but they typically incur significant computational cost during sampling. Moreover, direct learning of an inverse operator relies on high-fidelity datasets, which are often unavailable in real-world applications, further limiting the applicability of these approaches.

**First-Order Optimization and Its Challenges.** Gradient-based methods can efficiently navigate toward inverse solutions using differentiable surrogates. In principle, Neural Operators provide such surrogates Azizzadenesheli et al. (2024), enabling fast first-order optimization. However, FNOs are susceptible to adversarial examples, and naïve gradient descent often produces out-of-distribution solutions that are physically invalid. Constraining the search within a physically plausible manifold is therefore critical.

**Deep Generative Prior (DGP).** We propose **DGP**, a framework that combines a forward operator surrogate with a generative functional prior to constrain gradient-based inverse optimization. The key novelty lies in **learning the manifold of physically plausible solutions** directly from data and performing Langevin Dynamics (LD) in the latent space of the prior. This allows DGP to:

1. **Explore multiple valid inverse solutions**, capturing the inherent uncertainty of ill-posed problems.

2. **Mitigate adversarial examples**, ensuring physically plausible outputs even when the dataset contains suboptimal or noisy solutions.

3. Maintain **computational efficiency**, achieving orders-of-magnitude speedup compared to MCMC.

The **significance of using a generative FNO as a prior** is that it provides a **differentiable, data-driven manifold of valid solutions**, enabling robust exploration of the inverse solution space even when high-fidelity datasets or explicit physics-based priors are unavailable. Unlike directly learning an inverse operator, DGP allows **posterior sampling** over plausible solutions rather than producing a single deterministic point estimate, and it can refine solutions guided by the forward surrogate operator to improve physical consistency. We list the major contributions of this paper as follows:

- **FNO-Based Advantages:** DGP inherits FNO's strengths, including resolution invariance and free super-resolution, making it versatile for diverse inverse design tasks.

- **Purely Data-Driven:** DGP is fully data-driven, effective even when explicit physics models or priors are unavailable, intractable, or low-fidelity.

- **Diversity and Multi-Modal Solutions:** By sampling in the latent space of the generative prior, DGP can recover multiple physically plausible solutions for a single observation, capturing the inherent uncertainty of ill-posed problems and mitigating adversarial designs.

- **Superior Performance:** DGP demonstrates strong performance across a range of inverse design scenarios, including standard (Darcy Flow), under-determined or ill-posed (2D Navier-Stokes), and low-fidelity or out-of-distribution data (inverse lithography). It achieves comparable or better solution quality than existing methods while offering substantial computational speedup.

## 2 RELATED WORKS

**Fourier Neural Operator (FNO).** FNO Li et al. (2021) is a data-driven method for solving PDEs on discretized domains, performing iterative updates via integral kernel operators in the Fourier domain:

$$v_{t+1}(\boldsymbol{x}) = \sigma\big(\mathcal{F}_{\mathrm{op}}^{-1}(R_\phi \cdot (\mathcal{F}_{\mathrm{op}} v_t))(\boldsymbol{x}) + \mathcal{W} v_t(\boldsymbol{x})\big), \quad \forall \boldsymbol{x} \in D. \tag{1}$$

Here, $\mathcal{F}_{\mathrm{op}}$ is the Fourier Operator, $\sigma$ is a nonlinear activation, $\mathcal{W}$ a linear transform, and $R_\phi$ the Fourier kernel parameterized by a neural network. FNOs are resolution-agnostic and efficient as PDE solvers or surrogates, but they are sensitive to adversarial perturbations, which can hinder gradient-based inverse design.

**Deep Generative Models As Inverse Operator.** Generative models, including GANs Goodfellow et al. (2014a), GANOs Rahman et al. (2022), and diffusion-based models Wu et al. (2024); Ho et al. (2020), are used to model complex data distributions for inverse design. GANOs extend GANs to infinite-dimensional function spaces, generating functional data from Gaussian random fields. Diffusion models improve robustness to multimodal distributions and enable compositional design, though they are computationally intensive. All these models rely on training data quality and distribution, which can limit performance for out-of-distribution or low-fidelity datasets.

**Adversarial Sampling.** Deep models are vulnerable to adversarial examples due to discrete data and over-parameterization Goodfellow et al. (2014b). In inverse design, adversarial inputs can lead gradient-based optimization astray. LADA Liu et al. (2023) demonstrates generating functional adversarial designs with GANs that fool pretrained surrogate models while preserving realistic structure. This motivates constraining optimization with generative priors to maintain physical plausibility.

**Physics-Informed Neural Networks (PINNs).** PINNs Raissi et al. (2019) embed PDEs into the learning process, producing solutions that satisfy physical laws. They are interpretable and generalizable but can suffer from convergence issues, sensitivity to incomplete equations, and high computational cost. Our approach complements PINNs by using data-driven surrogates and generative priors to enable scalable, physically plausible inverse design.

**Connections to Deep Generative Prior (DGP).** While FNO provides a differentiable surrogate for the forward operator, deep generative models learn a data-driven manifold of physically plausible solutions. Diffusion or GAN-based generative models, as well as adversarial sampling studies, highlight the need to constrain inverse optimization to valid solution spaces to avoid unphysical or adversarial outputs. Our DGP framework integrates these insights by performing Langevin Dynamics in the latent space of a generative prior, guided by the differentiable forward surrogate. This combination enables efficient, physically plausible exploration of multiple inverse solutions, even under ill-posed or low-fidelity data conditions.

## 3 THE METHODOLOGY

### 3.1 PROBLEM FORMULATION

**Forward Operator.** Let $\mathcal{A}$ denote the input function space and $\mathcal{U}$ denote the output function space. The physical forward operator $\mathcal{F} : \mathcal{A} \to \mathcal{U}$ maps any input function $a : \mathcal{D}_{\mathcal{A}} \to \mathbb{R}^n$ to its corresponding output response $u = \mathcal{F}(a)$, where $u : \mathcal{D}_{\mathcal{U}} \to \mathbb{R}^{n'}$. Examples include partial differential equation (PDE) solvers, electromagnetic field simulators, or lithographic process simulators. Since $\mathcal{F}$ is typically nonlinear and computationally expensive, we adopt a data-driven surrogate operator. In particular, we train a Fourier Neural Operator (FNO) $F_\theta$ using supervised pairs $(a, u)$ sampled from $\mathcal{P}_{\mathcal{A}}$, where $a$ is an input function and $u$ is the corresponding solution. The learned surrogate satisfies $F_\theta(a) \approx \mathcal{F}(a)$.

**Inverse Design.** The inverse problem is defined as finding an input $a \in \mathcal{A}$ such that the forward response matches a target output $u^*$, i.e.,

$$a^* = \arg\min_{a \in \mathcal{A}} \|\mathcal{F}(a) - u^*\|^2. \tag{2}$$

This optimization is generally *ill-posed* and underdetermined because many different inputs may lead to similar outputs, especially for nonlinear PDE-based operators. We propose to address this challenge by: (1) replacing the expensive $\mathcal{F}$ with the learned surrogate $F_\theta$, and (2) incorporating a generative prior that captures the distribution of physically valid inputs.

### 3.2 ADVERSARIAL BEHAVIOR OF $F_\theta$ IN INVERSE DESIGN

When using the surrogate for inverse design, the natural optimization becomes

$$\hat{a} = \arg\min_{a \in \mathcal{A}} \|F_\theta(a) - u^*\|^2. \tag{3}$$

Expanding the error relative to the true operator $\mathcal{F}$, we obtain

$$\|\mathcal{F}(a) - u^*\|^2 = \|F_\theta(a) + \Delta(a) - u^*\|^2$$
$$= \underbrace{\|F_\theta(a) - u^*\|^2}_{\text{Surrogate Optimization Term}} + \underbrace{2\langle F_\theta(a) - u^*, \Delta(a)\rangle + \|\Delta(a)\|^2}_{\text{Residual Error}}, \tag{4}$$

where $\Delta(a) = \mathcal{F}(a) - F_\theta(a)$ is the surrogate error. Minimizing $\|F_\theta(a) - u^*\|^2$ is therefore not equivalent to minimizing $\|\mathcal{F}(a) - u^*\|^2$, unless $\Delta(a)$ is negligible. In practice, $\Delta(a)$ can be large when $a$ lies outside the training distribution $\text{supp}(\mathcal{P}_{\mathcal{A}})$. This phenomenon explains the existence of *adversarial designs*, i.e., input functions $a$ that minimize the surrogate loss but yield highly suboptimal true responses. We empirically verify this effect in results, where naive maximum likelihood estimation (MLE) yields unstable and inaccurate solutions.

### 3.3 DEEP GENERATIVE PRIOR

To address the limitations of directly optimizing over $F_\theta$, we define a conditional generative operator $\mathcal{G}$ that maps a Gaussian random input $q \sim \mathcal{N}(0, I)$ and a conditioning observation $u$ to a candidate design $a = \mathcal{G}(q, u) \in \mathcal{A}$. In this formulation, $\mathcal{G}$ is trained to model the conditional distribution $\mathcal{P}_{\mathcal{A}}(\cdot \mid u)$ of valid designs given the desired response $u$.

We adopt a conditional extension of the GANO framework, where the discriminator $d$ receives both $(a, u)$ pairs and enforces distributional alignment. The training objective follows the relaxed dual representation of the Wasserstein distance in function spaces:

$$\min_{\mathcal{G}} \max_{d} \mathbb{E}_{(a,u)\sim\mathcal{P}}[d(a, u)] - \mathbb{E}_{q\sim\mathcal{N}, u\sim\mathcal{P}_{\mathcal{U}}}[d(\mathcal{G}(q, u), u)] + \lambda \mathbb{E}_{(a,u)\sim\mathcal{P}'}\left[(\|\partial d(a, u)\|_{\mathcal{A}^*} - 1)^2\right], \tag{5}$$

where $d$ is a discriminator neural functional $d : \mathcal{A} \to \mathbb{R}$, $\mathcal{P}$ denotes the joint data distribution over $(a, u)$ pairs, and $\mathcal{P}'$ is the interpolation distribution $r\mathcal{G}_\# \mathcal{P}_{\mathcal{A}} + (1 - r)\mathcal{P}_{\mathcal{A}}$ used in the gradient penalty. Similarly we assume a discretized approximation of $\mathcal{G}$ as $G_\phi$.

**Posterior Sampling.** At inference time, given a new target $u^*$, the generator provides candidate designs $a = \mathcal{G}(q, u^*)$ consistent with the data-driven prior. To refine these candidates toward posterior consistency with the forward surrogate, we run Langevin dynamics in the latent space $q$:

$$q \leftarrow q - \gamma \, \partial_q \Big( \|u^* - F_\theta(\mathcal{G}(q, u^*))\|^2 + \lambda \|q\|^2 \Big) + \sqrt{2\gamma} \, \mathcal{N}, \tag{6}$$

where $\gamma$ is the step size. This procedure enforces agreement between the conditional generator and the differentiable surrogate model, producing samples that approximate the posterior distribution of designs given $u^*$. Note that without the noise term, Equation (6) reduces to gradient descent with $\ell_2$ regularization for maximum a posteriori (MAP) estimation.

## 3.4 DISCUSSION

**Inverse Design Error.** We measure the inverse design quality via

$$\mathcal{L}(a) = \|\mathcal{F}(a) - u^*\|. \tag{7}$$

Let $a^*$ be the optimal solution in $\mathcal{A}$, i.e., $a^* = \arg\min_{a \in \mathcal{A}} \mathcal{L}(a)$. Let $\hat{a} = G_\phi(q^*)$ be the solution obtained by our framework. We establish the following bound.

**Lemma 1.** *Assume: (1) $G_\phi$ can approximate $a^*$ within error $\epsilon_G$, i.e., $\|G_\phi(q^*) - a^*\| \leq \epsilon_G$ for some $q^*$; (2) the surrogate $F_\theta$ is $L_F$-Lipschitz; (3) the surrogate approximates the true forward operator within error $\epsilon_F$ on the generator range, i.e., $\|\mathcal{F}(a) - F_\theta(a)\| \leq \epsilon_F$ for all $a \in range(G_\phi)$.*

*Then the inverse design quality is bounded as*

$$\mathcal{L}(\hat{a}) \leq \mathcal{L}(a^*) + L_F \epsilon_G + 2\epsilon_F. \tag{8}$$

*Proof.*

$$\begin{aligned}
\mathcal{L}(\hat{a}) &= \|\mathcal{F}(\hat{a}) - u^*\| \\
&= \|\mathcal{F}(\hat{a}) - F_\theta(\hat{a}) + F_\theta(\hat{a}) - F_\theta(a^*) + F_\theta(a^*) - \mathcal{F}(a^*) + \mathcal{F}(a^*) - u^*\| \\
&\leq \|\mathcal{F}(\hat{a}) - F_\theta(\hat{a})\| + \|F_\theta(\hat{a}) - F_\theta(a^*)\| + \|F_\theta(a^*) - \mathcal{F}(a^*)\| + \|\mathcal{F}(a^*) - u^*\|. \quad (9)
\end{aligned}$$

Considering the assumptions we have,

$$\|\mathcal{F}(a) - F_\theta(a)\| \leq \epsilon_F, \forall a, \tag{10}$$

and

$$\|F_\theta(\hat{a}) - F_\theta(a^*)\| \leq L_F \|\hat{a} - a^*\| \leq L_F \epsilon_G, \tag{11}$$

which yields

$$\mathcal{L}(\hat{a}) \leq \mathcal{L}(a^*) + L_F \epsilon_G + 2\epsilon_F. \tag{12}$$

$\square$

Lemma 1 provides actionable design guidance: (i) improving the expressiveness of $G_\phi$ reduces $\epsilon_G$ and ensures access to near-optimal designs, (ii) enhancing the generalization of $F_\theta$ reduces $\epsilon_F$, and (iii) joint improvement directly reduces the practical inverse design gap between our method and the golden solution obtained by the true forward operator.

**Ill-Posed Optimization.** Inverse design problems are often ill-posed: for a given observation $u^*$, there may exist multiple distinct designs $a \in \mathcal{A}$ such that $\mathcal{F}(a) \approx u^*$. Our framework naturally accommodates this setting by running Langevin dynamics (LD) in the latent space of the conditional prior. Rather than converging to a single point estimate, LD generates multiple valid solutions that are all consistent with the target $u^*$, providing a principled way to capture design diversity and uncertainty.

**Out-of-Distribution Optimization.** Another important setting arises when the target $u^*$ lies outside the distribution of the training data. In such cases, our method combines the conditional prior $\mathcal{G}$ with the differentiable surrogate $F_\theta$ to enable controlled exploration. The soft $\ell_2$ regularization in the Langevin update (Equation (6)) balances faithfulness to the surrogate physics with flexibility to move beyond the strict data manifold. This allows DGP to generalize effectively to OOD targets, as demonstrated in our lithography experiments with weak and noisy datasets.

Table 1: Inverse Design on 2D Darcy Flow.

| Method | Continuous | | Clipped | | Throughput (s) |
|---|---|---|---|---|---|
| | Rel Error | Max Error | Rel Error | Max Error | |
| GANO | 0.037 | 0.507 | 0.035 | 0.087 | 0.003 |
| DDPM (1000-step) | 0.587 | 121 | 0.227 | 0.315 | 13.63 |
| MCMC (w/ $F_\theta, G_\phi$) | 0.038 | 0.477 | 0.011 | 0.042 | 179 |
| LD (w/o $G_\phi$, random) | 0.876 | 757 | 0.052 | 0.133 | 22.7 |
| LD (w/o $G_\phi$, condition) | 0.134 | 9.416 | 0.013 | 0.058 | 22.7 |
| LD (w/ $G_\phi$, ours) | **0.023** | **0.236** | **0.011** | **0.034** | 9.16 |

## 4 EXPERIMENTS

In this section, we compare our deep physics prior methodology with several representative inverse design solutions on 2D Darcy Flow (**standard PDE**), Naiver-Stokes Flow (**Ill-Posed Problem**) and Inverse Lithography (**Ill-Posed** and **Out-of-Distribution Solution**). All experiments are conducted on a single NVIDIA RTX A6000 Ada with 48GB memory. Dataset and model details are provided in Appendix A for reproducing numerical results.

### 4.1 DARCY FLOW

We first evaluate DGP on the Darcy flow equation, a canonical elliptic PDE describing porous media transport:

$$-\nabla \cdot (a(x)\nabla u(x)) = f(x), \quad x \in \Omega, \tag{13}$$

$$u(x) = 0, \quad x \in \partial\Omega, \tag{14}$$

where $a(x)$ denotes the permeability field, $u(x)$ the pressure field, and $f(x)$ the source term. The inverse task is to recover $a(x)$ given observations of $u(x)$.

**Results.** Table 1 presents the inverse design results on the 2D Darcy Flow. The columns "Continuous" and "Clipped" correspond to two dataset variants based on different choices of $\psi$ to make the PDE elliptic (see Equations (22) and (23)). The relative error ("Rel Error") and maximum error ("Max Error") are computed between the predicted pressure field $\boldsymbol{U}$ and the ground truth $\boldsymbol{U}^*$. We compare five baseline methods with our proposed approach. "GANO" refers to a generative adversarial neural operator Rahman et al. (2022) trained as an inverse operator. "MCMC (w/ $F_\theta, G_\phi$)" employs Markov-Chain Monte Carlo using the surrogate forward model $F_\theta$ and prior $G_\phi$, combined with the No-U-Turn sampler Hoffman et al. (2014). "DDPM" refers the baseline diffusion model that becomes popular solving inverse designs Ho et al. (2020). "LD (w/o $G_\phi$, random)" and "LD (w/o $G_\phi$, condition)" are gradient-based optimization methods that descend through running Langevin Dynamics from $F_\theta$ without generative prior $G_\phi$ during optimization; the former initializes inverse design randomly, while the latter uses the sample from $G_\phi$. "LD (w/ $G_\phi$, ours) " is our method, which incorporates a deep generative prior into the optimization process. Our method achieves the lowest Rel Error and Max Error among all baselines. Notably, our method reaches accuracy comparable to MCMC while offering a $20\times$ speedup. Visualization examples are also available in Figure 2, where we can observe without prior model, first order inverse optimization will likely fail due to adversarial examples (see LD (w/o $G_\phi$)). Also, direct inverse operator from generative model (GANO, DDPM) tries to capture the mixed distribution from the entire challenging training dataset, yieding a sub-optimal output.

### 4.2 NAVIER-STOKES FLOW

We next evaluate DGP on the 2D incompressible Navier–Stokes equations in vorticity form:

$$\partial_t w(x,t) + u(x,t) \cdot \nabla w(x,t) = \nu\Delta w(x,t) + f(x), \quad x \in \Omega, t \in [0,T], \tag{15}$$

$$\nabla \cdot u(x,t) = 0, \quad x \in \Omega, t \in [0,T], \tag{16}$$

$$w(x,0) = w_0(x), \quad x \in \Omega, \tag{17}$$

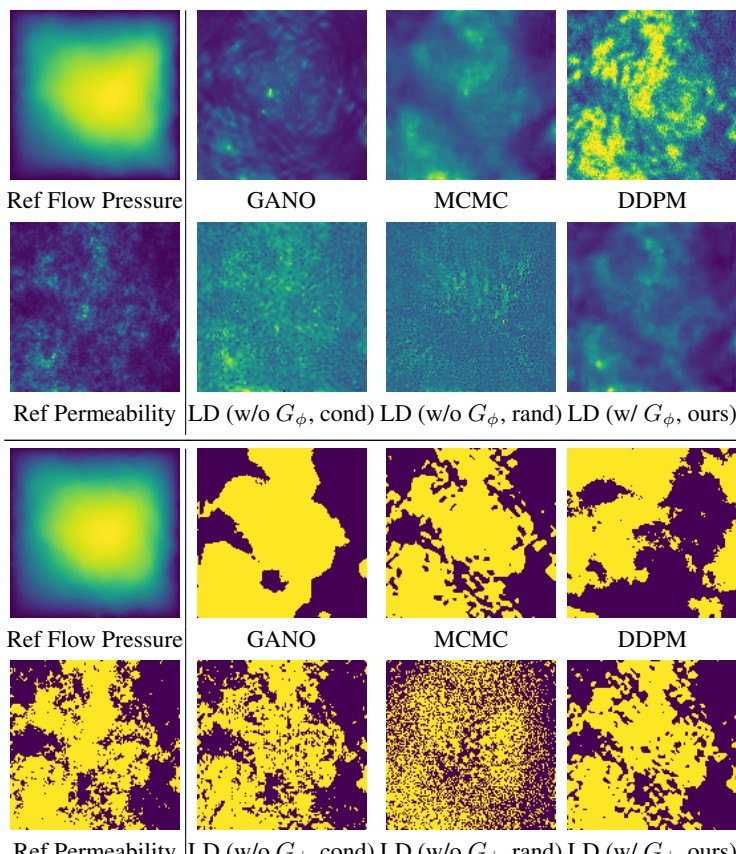

Figure 2: Visualization of inverse Darcy flow with exponentiated permeability (top) and clipped permeability (bottom). Left column: ground-truth reference flow pressure and permeability. Right columns: baseline methods and our deep generative prior with LD posterior sampling.

where $w(x, t)$ denotes the vorticity field, $u(x, t)$ the divergence-free velocity field, $\nu$ the viscosity, $f(x)$ the forcing term, and $w_0(x)$ the initial condition. The inverse task is to recover $w_0(x)$ given noisy or partial observations of $w(x, t)$ over time. Specifically, to make the inverse problem ill-posed, we intentionally set a larger viscosity $\nu = 1e{-}2$.

**Results.** Table 2 summarizes the inverse design performance on the Navier–Stokes problem. We evaluate reconstruction accuracy at the final timestep $T$, comparing the predicted vorticity trajectory $\{w_t\}$ against the ground-truth solution $\{w_t^*\}$ in terms of relative $L_2$ error and maximum error. The baseline methods follow the same setup as in Section 4.1: direct inverse models (GANO, DDPM), and

Table 2: Inverse Design on 2D Navier-Stokes Flow.

| Method | Rel Error | Max Error | Throughput (s) |
|---|---|---|---|
| MCMC | 0.047 | 0.049 | 43.04 |
| GANO | 0.059 | 0.050 | 0.003 |
| DDPM (1000-step) | 0.569 | 0.641 | 60.50 |
| LD (w/ $G_\phi$, ours) | **0.047** | **0.045** | 2.05 |

MCMC sampling with surrogate/prior $(F_\theta, G_\phi)$. Again, our method (LD w/ $G_\phi$) achieves the most accurate recovery across metrics, nearly matching MCMC while being substantially faster. Visual results are shown in Figure 3. Direct inverse operators (GANO, DDPM) produce low fidelity outputs that fail to capture small-scale vortical structures. In contrast, our DGP method preserves coherent flow structures while remaining consistent with the physical dynamics imposed by the surrogate model.

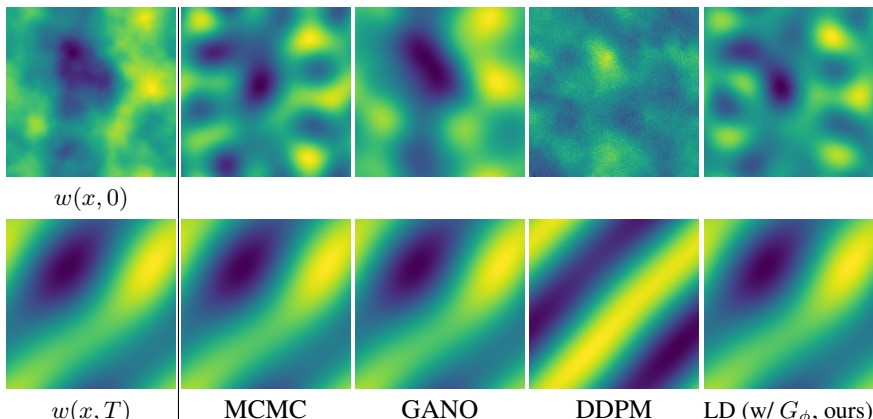

$w(x,0)$

$w(x,T)$ | MCMC | GANO | DDPM | LD (w/ $G_\phi$, ours)

Figure 3: Visualization of inverse Navier-Stokes flow on vorticity field. Left column: ground-truth reference vorticity at time step 0 ($w(x,0)$) and time step T ($w(x,T)$). Right columns: baseline methods and our deep generative prior with LD posterior sampling.

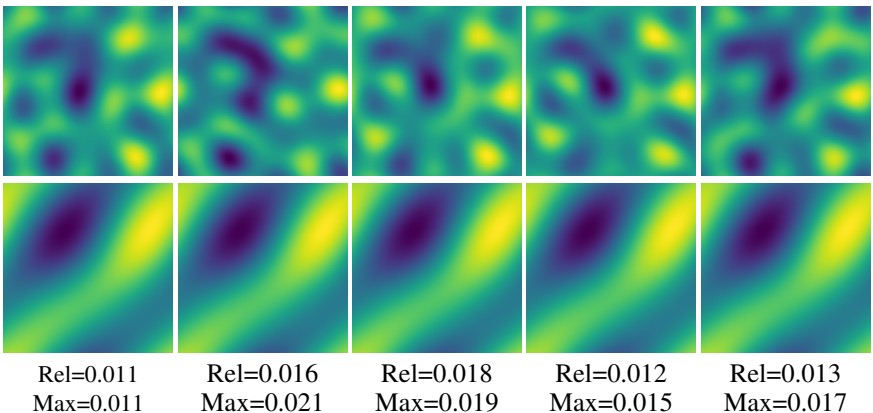

Rel=0.011 Max=0.011 | Rel=0.016 Max=0.021 | Rel=0.018 Max=0.019 | Rel=0.012 Max=0.015 | Rel=0.013 Max=0.017

Figure 4: Visualization of posterior sampling of five inverse solutions under ill-posed settings. Top: sampled $w(x,0)$; Bottom: solver derived $w(x,T)$.

**Solutions to Ill-Posed Problems.** In the Navier–Stokes experiments, we consider a relatively large viscosity $\nu$, under which different initial vorticity fields $w_0$ may evolve to nearly indistinguishable states $w(x,T)$ after a period of time. This creates an inherently ill-posed inverse setting, where multiple initial conditions are consistent with the same observations. To demonstrate the flexibility of our approach, we perform posterior sampling within the DGP framework, which efficiently explores diverse yet plausible initial states that reproduce the target solution. This highlights the capability of DGP to provide meaningful uncertainty characterization and support underdetermined inverse problems, as shown in Figure 4.

### 4.3 INVERSE LITHOGRAPHY

Lithography is the critical step to transfer chip physical designs onto silicon wafers. We model the forward lithography process using Abbe's partially coherent imaging formulation. The wafer intensity is

$$I(x,y) = \iint S(s_x, s_y) \left| \iint E\big(k_{\text{out}}; k_{\text{in}}(s_x, s_y)\big) P(p_x, p_y) \, e^{i(p_x x + p_y y)} \, dp_x dp_y \right|^2 ds_x ds_y, \tag{18}$$

where $S$ is the optical source distribution, $P$ the pupil function, and $E$ the diffracted field (requires EM simulation) depending on the mask $M$. Resist development is approximated by Gaussian blur

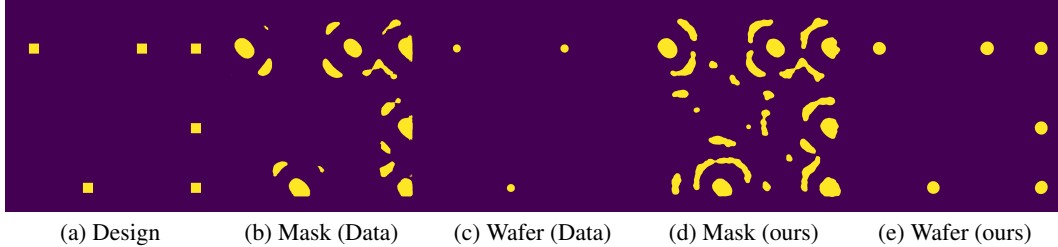

| (a) Design | (b) Mask (Data) | (c) Wafer (Data) | (d) Mask (ours) | (e) Wafer (ours) |

Figure 5: Visualization of inverse lithography solutions. (a) Snippet of the chip design. (b) The suboptimal mask of (a) in the dataset using numerical solver. (c) The wafer image of (b). (d) The optimized mask using LD. (e) The wafer image of (d).

and thresholding:

$$R(x,y) = (I * G_\sigma)(x,y), \qquad Z(x,y) = \mathbf{1}\{R(x,y) > \tau\}. \tag{19}$$

This defines the forward operator $\mathcal{F} : M \mapsto Z$. The inverse lithography task is

$$M^* = \arg\min_M \mathcal{L}\big(\mathcal{F}(M), Z^*\big), \tag{20}$$

with $\mathcal{L}$ measuring pattern fidelity (e.g., edge-placement error, the difference between the wafer patten and the original chip physical design). *It is important to note that practical inverse lithography depends on heuristics and approximations. As a result, high-fidelity datasets are rarely available, and the true optimal mask often lies outside the training distribution (OOD).*

**Results.** Table 3 summarizes the performance of different inverse lithography approaches on representative benchmarks. Numerical Solver Sun et al. (2023) achieves moderate EPE but is both slow and prone to local minima due to the highly non-convex nature of the optimization. Learned direct inverse methods, including ILILT and GANO, are trained on low-fidelity data, which makes it difficult

Table 3: EPE violation and throughput (patterns/sec).

| Method | EPE Violation | Throughput |
|---|---|---|
| Numerical Solver | 0.21 | 4.0 |
| ILILT-130K | 0.25 | 0.4 |
| ILILT-45M | 0.08 | 0.8 |
| GANO | 0.51 | 0.06 |
| **LD (w/ $G_\phi$, ours)** | **0.007** | 0.4 |

for the models to reach the true optimal mask; as a result, they either produce inaccurate masks or fail to capture fine pattern details. In contrast, our approach (LD w/ $G_\phi$) performs exploration on the mask manifold, leveraging the generative prior to guide optimization. This enables our method to achieve the lowest EPE violation (0.007), producing masks with high fidelity, while maintaining competitive throughput. Example results in Figure 5 show that LD masks closely match the intended patterns with minimal artifacts, demonstrating the effectiveness of combining generative modeling with inverse design exploration for *out-of-distribution solutions*.

## 5 CONCLUSION AND DISCUSSION

In this paper, we propose **Deep Generative Prior**, a data-driven methodology for first-order inverse design optimization. By jointly learning a forward surrogate operator $F_\theta$ that approximates the physical forward mapping $\mathcal{F} : \mathcal{A} \to \mathcal{U}$, and a generative model $G_\phi$ that maps latent variables $q$ to the design space $\mathcal{A}$, we enable differentiable, efficient, and physically consistent inverse optimization. We validate our approach on three representative case studies: 2D Darcy flow (**standard PDE**), 2D Navier-Stokes flow (**Ill-Posed**), and inverse lithography (**Ill-Posed** and **out-of-distribution solutions**). Across all cases, our method produces higher-quality solutions and demonstrates greater robustness compared to existing state-of-the-art techniques. The performance of DGP relies on the quality of both forward and generative operators which pose a trade-off of the optimization efficiency and performance. Future directions include case studies on more use cases and enabling deep physics prior on unstructured designs (e.g. graphs) to broaden the application scenario of this methodology.

**Reproducibility Statement**  While the code for our experiments is not publicly available at this time due to ongoing internal approval processes, we provide detailed descriptions of our data generation procedures, model architectures, and training protocols in the Appendix A. These details include dataset preparation, model hyperparameters, and training schedules, which are sufficient for reproducing the numerical results reported. We are committed to releasing the code once approvals are granted, ensuring full reproducibility of our methods.

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

## A  DETAILS OF DATA GENERATION AND MODEL TRAINING

### A.1  DARCY FLOW

**Dataset Generation.**  The Darcy flow equation is given in Eq. (14) of the main text. We construct datasets by sampling the permeability field $u(x)$ from a Gaussian random field (GRF) distribution:

$$\boldsymbol{A} \sim \psi_{\#}\mathcal{N}(\boldsymbol{0}, (-\Delta + \tau^2 \boldsymbol{I})^{-\alpha}), \tag{21}$$

where $\Delta$ is the Laplacian with Neumann boundary conditions. Following Li et al. (2021), the transformation $\psi$ enforces ellipticity of the coefficients. We consider both the clipped form and the exponentiated form.

- A clipping function Li et al. (2021):

$$\psi(x) = \begin{cases} 12, & \text{if } x \geq 0, \\ 4, & \text{otherwise.} \end{cases} \tag{22}$$

- An exponentiation function Rahman et al. (2022):

$$\psi(x) = e^x. \tag{23}$$

Unlike Li et al. (2021), where the hyperparameters $(\tau, \alpha)$ are fixed, we introduce variability to increase distributional diversity and hence challenging the task. Specifically, we sample

$$\alpha \sim \mathcal{U}(1, 2.5), \qquad \tau \sim \mathcal{U}(0.5, 1.5).$$

This produces permeability fields with heterogeneous correlation structures. We generate 6000 samples for training the surrogate and prior models and an additional 100 test samples for inverse optimization. We are focusing on a spatial resolution of $128 \times 128$ across the darcy flow experiments. Thanks to the properties of FNO, our flow extends to any resolution discretization.

**Model Architecture.**  Both the surrogate operator $F(\cdot; \theta)$ and the generative prior $G(\cdot; \phi)$ are implemented as Fourier Neural Operators (FNOs) with:

- Four Fourier layers,
- 32 feature channels,
- Maximum of 25 truncated Fourier modes.

**Training Setup.**  Both $F$ and $G$ are trained for 50 epochs using the `Adam` optimizer with an initial learning rate of 0.001 and cosine annealing scheduler. During inverse optimization, the surrogate and prior weights $(\theta, \phi)$ are frozen. Optimization is performed with the `Prodigy` optimizer Mishchenko & Defazio (2023), with a maximum of 100 iterations at fixed learning rate 1.0. The expected target pressure field is denoted as $\boldsymbol{U}^*$.

### A.2  NAVIER-STOKES FLOW

**Dataset.**  The initial condition $w_0(x)$ is sampled from a Gaussian random field (GRF) distribution with spectral density

$$w_0 \sim \mathcal{N}\big(0, (-\Delta + \tau^2 I)^{-\alpha}\big), \tag{24}$$

where $\alpha$ and $\tau$ control the smoothness and correlation length of the field. Unlike standard benchmarks with fixed parameters, we introduce variability by sampling $\alpha \sim \mathcal{U}(2, 2.5)$ while fixing $\tau = 7$, thereby generating diverse initial states.

The forcing function is defined as

$$f(x, y) = 0.1\big(\sin(2\pi(x + y)) + \cos(2\pi(x + y))\big). \tag{25}$$

We set the viscosity to $\nu = 10^{-2}$ and the final simulation time $T = 2.0$. Each trajectory is integrated using a pseudo-spectral solver with dealiasing and a Crank–Nicolson time-stepping scheme. We use an internal solver time step of $\Delta t = 10^{-3}$ and record 10 equally spaced snapshots in time.

The test dataset consists of 100 trajectories at a spatial resolution of $256 \times 256$. For each trajectory, we store the initial vorticity field $w_0(x)$ and the temporal sequence $\{w(x, t_j)\}_{j=1}^{10}$.

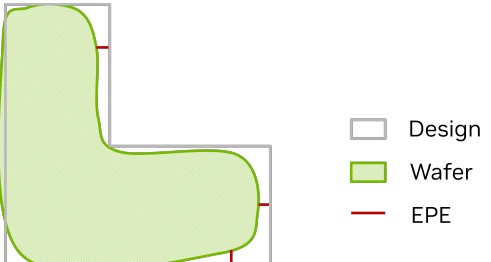

Figure 6: Measurement of inverse lithography performance using edge placement error.

**Model Architecture and Training.**    We adopt the same Fourier Neural Operator (FNO) backbone as in the Darcy experiments, using four Fourier layers with 32 channels and 25 truncated modes. The surrogate forward model $F(\cdot; \boldsymbol{w}_s)$ and the generative prior $G(\cdot; \boldsymbol{w}_p)$ are trained for 50 epochs with cosine annealing and a maximum learning rate of 0.001. Inverse optimization is performed with the `Prodigy` optimizer Mishchenko & Defazio (2023), up to a maximum of 100 iterations.

This setup evaluates DGP in an *ill-posed regime*, since different initial states $w_0$ can evolve to similar final states $w(x, T)$ due to dissipative diffusion. This makes the problem particularly challenging compared to Darcy flow.

### A.3    INVERSE LITHOGRAPHY

**Dataset.**    We adopt `LithoBench` Zheng et al. (2023) as our benchmark. The training set contains over 100K triplets of target layouts $\boldsymbol{Z}^*$, optimized masks $\boldsymbol{M}$ (obtained from numerical ILT solvers), and corresponding simulated intensity images $\boldsymbol{I}$ generated by the optical and resist models in Equations (18) and (19). In addition, `LithoBench` provides real-world layouts from the `Nangate45` standard cell library in the OpenROAD flow Ajayi et al. (2019), which we use to evaluate the generalization ability of our inverse lithography solutions. Due to the high accuracy demand, for the lithography tasks, we are dealing with data at 2048×2048 resolution discretized over $2\mu m \times 2\mu m$ chip area, posing additional challenges on resources and convergence.

**Model Architecture and Training.**    The forward surrogate $F_\theta$ is instantiated as a Fourier Neural Operator with 64 channels and 35 truncated Fourier modes, mapping masks to aerial images. For the prior model $G_\phi$, we adopt the convolutional FNO backbone from ILILT Yang & Ren (2024), which learns to generate plausible mask candidates given a target layout. Both $F$ and $G$ are trained with Adam optimizer under cosine annealing for 50 epochs with initial learning rate 0.001. During inverse optimization, $F$ and $G$ are frozen, and we perform Langevin dynamics in the mask space with Prodigy optimizer Mishchenko & Defazio (2023), using a maximum of 200 steps at fixed learning rate 1.0.

**Evaluation.**    We report Edge Placement Error (EPE) violations and throughput as our primary metrics (see Figure 6). EPE counts the number of locations where printed edges deviate from the target beyond a tolerance, while throughput measures runtime efficiency. These metrics allow a balanced comparison between numerical solvers, data-driven baselines, and our proposed DGP framework.

### A.4    BASELINE MODELS

**DDPM Baseline.**    As a baseline, we implement a denoising diffusion probabilistic model (DDPM) using a U-Net backbone from `diffusers` package. The model takes as input a 4-channel tensor, consisting of one channel for the permeability (or vorticity in NS2D) and three auxiliary channels for physical states and coordinates, and predicts a 1-channel update to the physical field. The U-Net consists of six hierarchical resolution levels, each with two convolutional layers per block. The channel widths across levels are $(128, 128, 256, 256, 512, 512)$, and attention is applied once in the down path and once in the up path at the bottleneck resolution. The downsampling is performed with `DownBlock2D` modules (including one `AttnDownBlock2D`), and the decoding path mirrors

this structure with `UpBlock2D` modules (including one `AttnUpBlock2D`). This design allows the baseline to combine local convolutional features with long-range attention, providing a fair and competitive comparison for PDE inverse modeling tasks such as Darcy flow and 2D Navier–Stokes.

**MCMC Baseline.** Across all tasks, we implement a Bayesian baseline using Markov Chain Monte Carlo (MCMC) with the No-U-Turn Sampler (NUTS) using `pyro`. The procedure introduces Gaussian perturbations on the observed output (e.g., $w(x, T)$ in PDEs or wafer image $\boldsymbol{Z}$ in lithography), which are processed by a pretrained inverse model $G$ to propose candidate inputs. These candidates are propagated through a pretrained forward surrogate $F$ to obtain predicted outputs. A Gaussian likelihood enforces consistency between predicted and observed outputs, yielding a posterior distribution over perturbations and corresponding candidate inputs. Posterior samples are drawn via NUTS, and the posterior mean is used for evaluation. This likelihood-based baseline provides a principled inference framework, in contrast to direct optimization or score-based methods.

## B  ADDITIONAL RESULTS

### B.1  TRAINING EFFORT.

Compared to generative approach or MCMC, our flow requires training of two networks. To provide a clear view of the computational cost associated with training our models, we summarize the training times for both the generative prior $G_\phi$ and forward surrogate $F_\theta$ across our case studies. All experiments were performed on a single RTX 6000 Ada GPU with 48 GB VRAM.

Table 4: Training time for generative prior ($G_\phi$) and surrogate ($F_\theta$).

| Case Study | $G_\phi$ Training | $F_\theta$ Training |
|---|---|---|
| Darcy 2D | 1 h | 1 h |
| Navier–Stokes 2D | 35 min | 35 min |
| Lithography | 12 h | 8 h |

Notes:

- Training is a one-time offline effort, enabling substantially faster and more stable inference during inverse design.
- Lithography training is longer due to the dataset size (over 100K training samples).
- WGAN (Wasserstein GAN) is used for generative model training to improve stability, mitigate mode collapse, and enhance generalization across varying data quality.

### B.2  STATISTICAL CONSISTENCY OF GENERATIVE MODELS.

To quantify the stability and reproducibility of inverse designs generated by our GAN-like models, we ran multiple experiments on inverse lithography using GANO with different random seeds. The following table reports the EPE violation on a subset of validation data:

Table 5: EPE violation for multiple random seeds.

| Seed | EPE Violation |
|---|---|
| 1 | 0.06 |
| 2 | 0.07 |
| 3 | 0.05 |
| 4 | 0.09 |
| 5 | 0.06 |

Observation:

- Results indicate consistent convergence across seeds, with low variance in solution quality.

- The training procedure, including WGAN stabilization, produces reliable inverse designs without significant mode collapse or instability.

## C   USE OF LARGE LANGUAGE MODELS

We note that a large language model (LLM) was used only to assist in polishing the wording, improving clarity, and formatting the paper. All technical content, ideas, experiments, and results were generated solely by the authors.

