# OpenReview forum: "Deep Generative Prior for First Order Inverse Optimization"
_ICLR.cc/2026/Conference — ICLR 2026 Conference Withdrawn Submission_

### Official Review · Reviewer_manJ · 2025-10-27

**Soundness:** 2
**Presentation:** 3
**Contribution:** 3
**Rating:** 6
**Confidence:** 3

**Summary:**

The paper aims to solve inverse problems with high throughput. This is done by training a generative operator that maps the observation to a distribution of action, and a discriminator that learns the joint distribution for (action, corresponding observation). The generative model can then be used to generate samples for the inverse problem solutions. The experiments show the use in different experiments including ill-posed and OOD cases.

**Strengths:**

The paper aims to tackle an important problem for inverse modelling, even if the problem setting is quite standard inverse problem. The angle about being able to do so at higher throughput is still of use nonetheless.

The proposed methods are described well, not too convoluted (seems to be from a standard method), and text is quite easy to follow.

The experiments are sufficiently complex, and the PDEs used in the setup are realistic. They also seem to be chosen quite well, to cover ill-posed conditions and out-of-distribution cases.

**Weaknesses:**

More information about how GANs were trained in this case should be mentioned, since they are difficult to train in practice

Sec 3.4 -- while the proof is correct, it is quite trivial (so may be okay to omit from main text), and the results it show is trivial. It just says "I train to get lower error = problem will also have lower error". It would be much more informative if, for example, the result can link the achieved error to the use of certain choice of NN architecture, or some changes to the GAN method proposed (if there were any).

Results have no error bars.

While the paper does mention the use of experiments on ill-posed problems, the metrics reported may not necessarily be the most suitable since it only compares the prediction against a point prediction, which ignores the ill-posedness of the solution. I feel whether some metric based on uncertainty would be better to respect that there can be many solutions that can match the same observations.   Would welcome suggestions or comments by the authors though.

The OOD case could probably be more convincing, by constructing a dataset where the test set is more explicitly OOD from the training set (e.g., generated from different priors).

[Minor formatting point] Abstract should be formatted into one paragraph

**Questions:**

1. In this case, why are GANs used? Could other pairs of models that can perform backward/forward for the same problem have also been used instead? Would like to hear the use of GANs (as opposed to other architectures) more motivated.

2. Can the proposed method capture the uncertainty of the predicted outcome, in the way that some of the other benchmarks such as MCMC can do? In the experiments, it is arguable that it is better than MCMC in practice since it seems to be constructed for a slightly different use case than MCMC.

3. Related, can the proposed method handle noisy inputs? This again would be another case that could be demonstrated to contrast with existing probabilistic generative methods.

---

### Official Review · Reviewer_XeTK · 2025-10-27

**Soundness:** 2
**Presentation:** 2
**Contribution:** 1
**Rating:** 0
**Confidence:** 5

**Summary:**

This work combines manifold learning with Langevin dynamics to solve inverse problems by sampling the posterior distribution, given a forward function (essentially modeling the likelihood) defined over an intrinsically low-dimensional prior. The method is then employed in several inverse problems like Darcy flow, NS flow and lithography.

**Strengths:**

1. Reasonable combination of existing ideas like GANO, FNO and LD.
2. Paper is clearly written.
3. The method is compared to several baselines and seems to perform better.

**Weaknesses:**

The limited novelty and experimental evaluation lead me to reject this paper:
1. Limited novelty. The idea is not new. For instance:
     * Nguyen, Thanh V., Gauri Jagatap, and Chinmay Hegde. "Provable compressed sensing with generative priors via Langevin dynamics." arXiv preprint arXiv:2102.12643 (2021).
     * Nguyen, Thanh V., Gauri Jagatap, and Chinmay Hegde. "Inverse imaging with generative priors via Langevin dynamics." ICASSP 2022-2022 IEEE International Conference on Acoustics, Speech and Signal Processing (ICASSP). IEEE, 2022.
     *  Coeurdoux, Florentin, Nicolas Dobigeon, and Pierre Chainais. "Normalizing flow sampling with Langevin dynamics in the latent space." Machine Learning 113.11 (2024): 8301-8326.
2. Not enough insights provided about the components and the choice of crucial hyperparameters. For instance:
    1. This paper assumes that the prior is supported by a low dimensional structure in the high dimensional space, and uses a conditional GANO $G(q, u)$ to approximate this structure as a prior. However, I think the $\epsilon_G$ in Lemma 1 is too crude to summarize all the errors of $G$. Many need careful investigation, as they may have great impact on solving inverse problems especially ill-posed ones. For instance:
        1. The authors didn't show how the dimensionality of the latent $q$ is chosen. This is a crucial parameter, as if it's set too small, the image of $G$ can at best only cover a negligible part of the true manifold structure supporting the data distribution. This can impair the model's performance in retrieving all valid solutions. See Lemma 1 of _"Arjovsky, Martin, and Léon Bottou. Towards principled methods for training generative adversarial networks. arXiv preprint arXiv:1701.04862 (2017)."_, or _Sard's theorem_ in differential topology for a more rigorous analysis.
        2. The conditional GAN $G(q, u)$ is not guaranteed to be injective w.r.t. $q$ with $u$ fixed, so for each $a$ there can exists multiple $q$ producing the same $a = G(q, u)$, and I expect the initialization of $q$ to be important to the solution, especially considering that you use Langevin dynamics to sample the latent space. This should be investigated and discussed,
     2. $\gamma$ in Langevin dynamics is a crucial hyperparameter. Its effect should be discussed. For instance, it is problem dependent? How to find an acceptable $\gamma$ for each problem? How does the design of $G$ affect the choice?
3. Some unsound parts, and limited experiments. For instance:
    1. _"DGP allows posterior sampling over plausible solutions rather than producing a single deterministic point estimate"_ on page 3 and _"Note that without the noise term, Equation (6) reduces to gradient descent with ℓ2regularization for maximum a posteriori (MAP) estimation"_ on page 5.
         * However, in the experiments, all I saw are Rel Error and Max Error for evaluating a single solution's agreement with the ground truth. Only a very small paragraph on page 8 is dedicated to ill-posed problem, and the investigation is very limited. I think more rigorous evidence is need to show its ability to handle ill-posed problems. For instance, how many valid solutions can it actually recover? Can it recover all?
         * The high dimensional problems are appealing but also sometimes misleading. To more intuitively investigate the model's ability to recover all (or practically most of) the solutions in ill-posed problem, I suggest the authors start with a simple 2D ill-posed inverse problem: Try a function $F: R^2 \to R$ with 2 peaks and a saddle point, like the product of two 2D Gaussian functions of different means. Set $\dim q$ to 1 or 2 for your $G$, and see if your model can recover all the solutions for each value of $F$.
    2. The claim that "This combination enables efficient, physically plausible exploration of multiple inverse solutions,
**even under ill-posed or low-fidelity data conditions**" on page 3.
          * More experiments and problems are needed to support this claim.
    3. For the OOD optimization, I think it’s necessary to report the error quantity on the RHS of Equation 20 (denoted as $M$) along with EPE. The number of test samples should also be provided. Both the EPE violation and the error $M$ are better presented as histograms rather than as single scalar values.
    4. When solving ill-posed inverse problems, we should use statistical divergences (or distances) to evaluate the target and approximate distributions' difference, apart from visual inspections or per-sample evaluations. The authors may use MMD as a metric for their application, for instance.

**Questions:**

1. Why not replace $G(q,u)$ with an unconditional $G(q)$? In my experience $G(q,u)$ is in general much harder to train accurately than $G(q)$ due to the topological complexity of $F$'s preimages (take the 2-peak-1-saddle 2D problem I mentioned above as an example), and I don't think removing the input condition $u$ in $G$ can make your algorithm any greatly slower. It probably would be more accurate imo.
2. I'm skeptical about the DDPM's bad accuracy when solving inverse problems. Could the authors provide some insights on why it's such bad?

---

> ### Author Response · Authors · 2025-11-12
> **request that the 0-score review be treated as an outlier and excluded from the meta-decision.**
>
> Dear AC,
>
> We respectfully request that the 0-score review be treated as an outlier and excluded from the meta-decision.
>
> Limited-novelty claim relies on keyword matching (“Langevin dynamics”) and cites works outside our scope; none address PDE-governed inverse design with neural-operator surrogates plus a conditional functional prior and latent-space LD as we do.
>
> Material misreadings: the review states we show only single-solution metrics, yet we include explicit multi-sample posterior results in the ill-posed setting; it recommends replacing G(q,u) with G(q) while not engaging our ill-posedness discussion where conditioning narrows the feasible set.
>
> Low-engagement nitpicks: asks for a toy R^2→R example unrelated to our PDE/lithography domains and gatekeeps with extra plots (γ sensitivity, histograms/MMD) while overlooking core analysis.
> A score of 0 is inconsistent with the manuscript’s clarity, analysis, and multi-domain results, and with the other two reviews (6, 6).
>
> We are happy to add the suggested ablations as normal improvements, but they are not grounds for a 0. We ask that this review be discounted or replaced by an additional expert review. A timely decision would help guide whether we remain with ICLR or pivot to another venue.
>
> Sincerely,
>
> The authors

---

### Official Review · Reviewer_ppaE · 2025-11-01

**Soundness:** 3
**Presentation:** 3
**Contribution:** 3
**Rating:** 6
**Confidence:** 4

**Summary:**

This paper proposes Deep Generative Prior (DGP), a framework for enabling first-order, gradient-based inverse optimization in systems lacking explicit mathematical formulations, such as those in fluid dynamics, materials science, and semiconductor manufacturing. DGP constrains design parameter optimization using a pretrained generative prior G(q), with gradients propagated through a surrogate forward model F (e.g., Fourier Neural Operator). This approach mitigates issues like adversarial examples and ill-posedness by optimizing along a learned data manifold. The method is evaluated on 2D Darcy flow (standard), 2D Navier-Stokes (ill-posed), and inverse lithography (ill-posed with out-of-distribution solutions), demonstrating improved solution quality, diversity, and efficiency over baselines like GANO, DDPM, MCMC, and direct Langevin dynamics.

**Strengths:**

* Tackles a practical and timely problem in inverse design by combining generative priors with differentiable surrogates, enabling efficient optimization in data-driven settings without relying on high-fidelity paired data or explicit priors.

* Provides a clear theoretical analysis, including an error bound that links surrogate approximation, generator expressiveness, and Lipschitz constants, offering actionable insights for improving the framework.

* Empirical results are compelling, showing consistent advantages in error metrics and throughput across diverse tasks, including ill-posed and OOD scenarios, while capturing multi-modal solutions via latent-space Langevin dynamics.

* The fully data-driven nature and resolution invariance (inherited from FNO) make it versatile for real-world applications where physics models are intractable or low-fidelity.

**Weaknesses:**

* The evaluation is somewhat narrow, focusing primarily on PDE-based tasks in 2D; it's unclear how DGP scales to higher-dimensional problems, non-PDE systems, or domains with different data modalities (e.g., graphs or sequences), potentially limiting generalizability claims.

* While adversarial behavior of surrogates is discussed, the paper doesn't provide quantitative ablation on robustness (e.g., under varying noise levels or dataset shifts), and the reliance on FNO might inherit its known limitations like sensitivity to discretization.

* Reproducibility could be improved; key details like hyperparameter choices for Langevin dynamics (e.g., step size γ, noise scaling), exact dataset generation processes, and full training costs for G and F are not fully detailed in the provided excerpt, though appendices are mentioned.

* Comparisons to baselines like DDPM use a 1000-step sampler, which may unfairly highlight DGP's efficiency; faster DDPM variants or other diffusion schedulers could narrow the gap.

**Questions:**

* How sensitive is DGP to the choice of generative model architecture (e.g., GANO vs. other functional generators), and were alternatives explored?

* In the lithography experiments, how was out-of-distribution handled in dataset curation, and what specific metrics show DGP's superiority in low-fidelity settings?

* Does the method exhibit biases in multi-modal solution sampling (e.g., favoring certain modes), and how does it compare to pure MCMC in capturing posterior diversity?

---

### Note · Authors · 2025-11-18

**Comment:**

It's said we have irresponsible reviewer and conference does not take serious attention. Feel sorry about what iclr becomes.

**Withdrawal Confirmation:**

I have read and agree with the venue's withdrawal policy on behalf of myself and my co-authors.